# A Study on the Causes of Apomixis in *Malus shizongensis*

Yuchen Feng [1], Ruiyuan Ning [1], Zidun Wang [1], Ying He [1], Yu Hu [1], Lulong Sun [1] and Zhenzhong Liu [1,2,*]

[1]   College of Horticulture, Northwest A & F University, Xianyang 712100, China;
      fengyuchen0622@163.com (Y.F.); ningruiyuan@163.com (R.N.); w17835697813@163.com (Z.W.);
      heying01128@163.com (Y.H.); huyu5405@163.com (Y.H.); lulongsun@126.com (L.S.)
[2]   Apple Engineering and Technology Research Center of Shaanxi Province, Xianyang 712100, China
*     Correspondence: liuzhenzhong@tom.com; Tel.: +86-029-8708-2922

**Abstract:** Apomixis is a unique reproductive process that produces fertile offspring without the combination of sperm and egg cells. This process perfectly reproduces maternal DNA, making it possible to fix heterosis during reproduction. *Malus shizongensis* is a newly discovered species that is closely related to *Malus hupehensis* Rehd. After de-male bagging, it was found that the fruit set rate reached 78.7%. Preliminary analysis indicated that *M. shizongensis* have apomictic reproductive characteristics. In this work, we employed paraffin sectioning and electron scanning microscopy to explore apomixis in *M. shizongensis* during the development of male–female gametes and embryo sacs. Stigma fluorescence assays showed that pollen germination was normal, but less pollen entered the ovaries. Additionally, analysis of anthers indicated the presence of dysplasia and paraffin sectioning revealed that the pollen mother cells were aborted due to abnormal disintegration of the tapetum layer. Taken together, our results indicate that the primary causes of apomixis in *M. shizongensis* are anther dysplasia and male gamete development failure, resulting in reduced pollen tube entry into ovaries and reduced reproduction of female gametes. In conclusion, this study provide a theoretical basis and technical supports for apple stock breeding and apple industry development.

**Keywords:** *Malus shizongensis*; apomixis; embryo sac development; paraffin section; fluorescence detection





## 1. Introduction

Apomixis produces offspring without sperm–egg interaction and was first discovered by Smith in *Aconitum* [1], and have been described in more than 400 flower plant species. Thus far, the majority of research on apomixis has been conducted on *Hieracium umbellatum* L., *Tripsacum* L., *Arabidopsis thaliana* and other related wild species [2]. Additionally, breeding applications have been explored in Gramineae plants, such as *Oryza sativa* L. and *Pennisetum alopecuroides* L. Spreng [3,4]. Research on apomixis in *Malus* began in 1931, when Sax discovered the parthenogenesis of *M. hupehensis* Rehd. Since then, apomixis has been reported in *Malus sargentii* Rehd., *Malus sikkimensis* Koehne., *Malus sieboldii* Rehd., *Malus toringoides* Hughes., *Malus platycarpa* Rehd., *Malus rockii* Rehd., *Malus xiaojinensis*, *Malus coronaria* Mill., *Malus lancifolia* Rehd. and others [5]. Apomixis was found to occur in more than 400 plant species in 40 families, including 10 species and 23 varieties in *Malus* [6,7]. Apomixis can be divided into two types according to the origin and development process of embryos: sporophytic and gametophytic [8,9]. With respect to sporophytic apomixis, the embryo originates from somatic cells in ovules after mitosis. In sporophytic apomixis, the embryo sac is formed via meiosis and mitosis of the nucleus, which is differentiated from integument cells into adventitious embryos. Gametophytic apomixis can be further divided into diploid spore reproduction and apomixis, depending on the source of the embryo sac [10]. Diploid spore reproduction occurs through mitosis of megaspore mother cells into the embryo sac, which then directly develops into embryos through diploid egg cells. The apomixis of the sporophyte involves the formation of the embryo sac via mitosis

of nucellar cells, and the embryo is formed by the differentiation of asporogenous initial cells in the embryo sac [11].

*M. shizongensis*, a small deciduous tree belonging to Rosaceae, was introduced from Kunming, Yunnan Province in 2012 by researchers of the Northwest A&F University and planted in the Qingcheng Experimental Station in 2013. Through SNP analysis, it was confirmed as a new species of apple. Phylogenetic tree analysis showed that *M. shizongensis* is closely related to *M. hupehensis* Rehd., which is consistent with the analysis of its morphology. This same phylogenetic analysis indicated that it is distantly related to *Malus sieversii*, *Malus prunifolia* (Willd.) Borkh and *Malus sylvestris* Mill [12]. Therefore, *M. shizongensis* may have the same properties as *M. hupehensis* Rehd. Although *M. hupehensis* Rehd has been shown to undergo apomixis, this phenomenon has not been systematically studied in its close relative *M. shizongensis*. Previous work on apomixis in *M. xiaojinensis* has primarily focused on anatomical observation of male and female stamens [13]. This earlier work revealed that *M. xiaojinensis* male stamens aborted and failed to produce pollen grains. Without meiosis, the cytogenic cell directly undergoes three rounds of mitosis to form an embryo sac with seven nuclei and eight cells [13]. Previously, it was found that most embryo sacs of *Malus hupehensis* var. *pingyiensis* were abortive at full flowering stage, although a few did continue to develop. Earlier work has also shown that after pollination, the pollen tube of *M. hupehensis* var. *pingyiensis* detaches at the bottom of the style and only a small portion of the tube enters the ovary, which was one of the reasons for the apomixis of *M. hupehensis* var. *pingyiensis*. Female gametophytes develop within the ovule, often referred to as the embryo sac or gametophyte, while male gametophytes, usually called pollen grains or small gametophytes, develop within the anthers. The formation of male and female gametophytes is necessary for the development of sexual and asexual seeds in angiosperms, making an in-depth understanding of this process critical [14].

In this work, we examined the development of pistils, stamens and embryo sacs via paraffin sectioning, stigma fluorescence detection, and scanning electron microscopy to better understand the underlying causes of apomixis in this species. A better understanding of the causes behind *M. shizongensis* apomixis lays a foundation for subsequent breeding work. Make it a new germplasm resource in rootstock breeding to enrich the diversity of self-rooting anvil, and it can retain its dwarfing characteristics. Therefore, breeding is in line with the current development trend.

## 2. Materials and Methods

Plants used for this study were grown from April to November 2022 at Qingcheng Apple Experimental Station of the Northwest A&F University, Gansu Province, with an altitude of 1200–1600 m, a latitude of 35°42′29″–36°17′22″, an annual rainfall of 537.5 mm, an annual average temperature of 9.4 °C and a frost-free period of 150 days °C.

### 2.1. Experimental Design

Six-year-old *M. shizongensis* was used as the experimental material, which was grown with conventional fertilizer, water management and pest control methods. Three healthy plants were selected and subjected to three treatments of de-male pollination, de-male bagging and de-stigma bagging at the large bud stage (22 April). A total of 20–30 inflorescences were selected from each tree in each treatment, leaving a variable number of flowers at each inflorescence, depending on the current growth stage. All samples were collected with three biological replicates. Bags were removed ten days after initial treatment to determine the fruit setting rate, followed by a second examination one month later. Buds, flowers and young fruit were picked after inflorescence separation (7 April), while anthers were retained until a week after flowering (2 May). Ten samples were collected at each time point and immediately put into FAA (Formalin-Aceto-Alcohol) fixing solution, followed by storage at 4 °C. At the full flowering stage (25 April), 50 mature flowers were sampled. Anthers from these flowers were stripped away, allowed to dry naturally in a shaded area and stored in dry glass bottles for subsequent observation of three-dimensional anther structures.

### 2.2. Experimental Methods

### 2.2.1. Investigation of Apomixis Rate and Fruit Character

Three healthy plants were selected and subjected to three treatments of de-male pollination, de-male bagging and de-stigma bagging at the large bud stage (22 April). Each treatment was divided into three groups. After physiological fruit dropping (22 May), fruit setting and apomixis rates were calculated. After fruit ripening, the fruits of each treatment were collected separately, and the longitudinal and transverse diameters of the fruits were measured with electronic vernier calipers. The seeds were then removed and examined to calculate seed number and satiety rate, and the number of single fruit seeds and the number of thousand seeds were calculated.

The experimental data were measured with 3 biological replicates. Microsoft Office Excel 2010 was used for data processing and mapping, and SPSS 26.0 software (IBM company, New York, NY, USA) was used for data significance analysis. Data are expressed as mean ± standard error.

### 2.2.2. Stigma Acceptability Test

Column receptiveness was determined using the benzidine-hydrogen peroxide method of Ferreira, with minor modifications [15]. Briefly, the flowers were collected at 10 am and 4 pm, and the column heads were placed into concave slides. A benzidine-hydrogen peroxide reaction solution (1% benzidine: 3% hydrogen peroxide: water in a 4:11:22 volume ratio) was added and the glass was then covered and observed under a microscope. If the stigma was permeable, the reaction liquid around the stigma appeared to be blue and was full of bubbles. The receptivity of stigmas was then calculated based on the number and size of bubbles.

### 2.2.3. Stigma Fluorescence Detection

After successful artificial pollination, samples were taken for the next 72 h, at 12 h intervals. Fuji was used as a negative control. Fluorescence microscopy with UV excitation of light (BX51 positive fluorescence microscope +IX71 inverted fluorescence microscope, made in Tokyo, Japan) was used for observation, and Castro's method was adopted [16]. The specific methods were as follows:

(1) Stigmas 12 h after artificial pollination were selected and fixed in an FAA (Formalin-Aceto-Alcohol) fixation solution;
(2) Eight $mol \cdot L-1$ NaOH was softened at 75 °C for 15–20 min;
(3) Samples were rinsed with deionized water several times;
(4) A dye solution containing 0.1% aniline was applied to the sample for 4 h in dark conditions;
(5) One drop of deionized water was dropped on the slide and the stained stigma was placed on the slide, which was then covered. Ultraviolet excitation light was used as illumination under a research-grade microscope for observation and photography.

### 2.2.4. Observation of Embryo Sac Development

The paraffin sections were developed according to the Cerovi method [17], with minor modifications. Briefly, the material was fixed in FAA (formalin: propionic acid: 70% acetic acid in a 5:5:90 ratio) and stored at 4 °C. Samples were then dehydrated with a series of ethanol rinses and then formed into a paraffin block, which was sectioned with a Leica RM2155 microtome. The sections were fixed and attached to the slides, followed by staining with toluidine blue. Observations were carried out with a research-grade microscopic imaging system (BX51 positive fluorescence microscope +IX71 inverted fluorescence microscope).

### 2.2.5. Observation of Pollen Morphology and Structures

The anthers of 10 flowers were stripped off and mixed together for shade drying; 10 dried anthers were randomly selected and uniformly spread onto the loading platform with a double-sided conductive adhesive coating with tweezers and cotton sticks,

respectively. After spraying gold on the samples via the ion sputtering method, the anther and pollen of *M. shizongensis* were placed on the sample table. The temperature of the sample seat was 5 e, and the voltage was 5 KV. Pollen grains were selected for observation with scanning electron microscopy (ESEM, S-3400N, made in Tokyo, Japan) and then photographed.

2.2.6. Observation of Male Gamete Development

Flowers with normal growth were sampled and their petals and sepals were removed, leaving only the ovaries and stamens. The flowers were then fixed in an FAA solution, and air was pumped into the bottom of the bottle, which was then placed in a 4 °C refrigerator. Paraffin sectioning and preparation were carried out via the same methods used for embryo sac observation, as described previously by Cerovi [17].

## 3. Results

*3.1. Statistics of Fruit Setting Rate and Fruit Character of M. shizongensis under Different Treatments*

This can be obtained with significance analysis: the fruit setting rates of the three different treatments of *M. shizongensis* were found to be over 90% at seven days after the budding stage (Table 1). However, there were no significant differences between the three treatments. The fruit setting rate at 30 days after bud stage was lower than that at seven days. At this point, de-male bagging was significantly lower than de-male pollination bagging and de-stigma bagging, but there was no significant difference between de-male pollination bagging and de-stigma bagging. These results indicated that under all three treatments, *M. shizongensis* could maintain a high fruit setting rate. These data indicate that *M. shizongensis* can set fruit without pollination, which is a key characteristic of apomixis.

**Table 1.** Fruit setting rate in 7 and 30 days after pollination in buds in *M. shizongensis*.

| Pollination in Buds | Number of the Pollinated Buds | Number of Fruits in 7 Days after Pollination | Fruit Setting Rate (%) | Number of Fruits in 30 Days after Pollination | Fruit Setting Rate (%) |
|---|---|---|---|---|---|
| DMPB | 191.00 | 180.00 | 94.20 ± 2.72 a | 167.00 | 87.40 ± 2.65 a |
| DMB | 230.00 | 215.00 | 93.50 ± 1.51 a | 181.00 | 78.70 ± 3.03 b |
| DSB | 108.00 | 104.00 | 96.30 ± 1.73 a | 98.00 | 90.70 ± 3.04 a |

Different letters indicate significant differences at $p < 0.05$ (Duncan's Multiple Range Test). Fruit setting rate (%) = number of fruits in 7 (30) days after pollination/number of pollinated buds × 100%. DMPB stands for de-male pollination bagging, DMB stands for de-male bagging, and DSB stands for de-stigma bagging. Each treatment is divided into three groups. DMPB's three groups processed 62, 63 and 66 respectively, and the sum of the three groups is shown in the table. The three groups of DMB processed 81, 73 and 76, respectively. Similarly, the three groups of BSB processed 35, 33 and 30, respectively, and the sum of the three groups is shown in the table. In the table, the fruit setting rates of different treatments in 7 and 30 were also the sum of the three groups.

The longitudinal diameter of the castration bagging treatment was the largest, which was significantly different from the longitudinal diameter of de-stigma bagging (Table 2), but had no significant difference compared with the longitudinal diameter of castration pollination bagging. However, there was no significant difference in transverse diameter, fruit shape index or diameter/transverse diameter of the fruits across any of the three treatments, indicating that *M. shizongensis* fruits maintained a similar shape under all conditions. The 1000-seed weight of the de-pollination seeds was 6.66 g/ 1000 seed, which was higher than that of the other two treatments.

**Table 2.** Fruit character after pollination in buds in *M. shizongensis*.

| Pollination in Buds | Fruit Transverse Diameter (mm) | Fruit Longitudinal Diameter (mm) | Fruit Shape Index | Fruit Number | Seed Number | 1000-Seed Weight (g) |
|---|---|---|---|---|---|---|
| DMPB | 11.03 ± 0.61 a | 10.21 ± 0.50 ab | 1.08 ± 0.056 a | 167.00 | 613.00 | 6.66 ± 0.46 a |
| DMB | 11.32 ± 0.97 a | 10.59 ± 0.74 a | 1.07 ± 0.079 a | 181.00 | 644.00 | 5.29 ± 0.30 b |
| DSB | 11.03 ± 0.60 a | 9.89 ± 0.54 b | 1.12 ± 0.032 a | 98.00 | 340.00 | 5.34 ± 0.42 b |

Different letters indicate significant differences at $p < 0.05$ (Duncan's Multiple Range Test). Fruit shape index = fruit transverse diameter/fruit longitudinal diameter. DMPB stands for de-male pollination bagging, DMB stands for de-male bagging, and DSB stands for de-stigma bagging. Consistent with Table 1, the number of fruits and seeds under different treatments is the total.

### 3.2. Low Pollen Germination Rate and Sexual Embryo Sac Abortion in M. shizongensis

3.2.1. Stigma Acceptability Test

Previous studies have shown that increased levels of bubble generation around stigmas are correlated with the level of stigma acceptability [18]. As shown in Figure 1, there are more bubbles around 10 a.m (Figure 1A) than 4 p.m. (Figure 1B), so stigma receptivity is stronger at 10 a.m. than at 4 p.m. The results showed that the receptivity of stigmata to pollen was strong and the pollination success rate was high in the morning pollination, and pollen germination in stigma will not be reduced because of the stigma, so the fruit setting rate will not be affected.

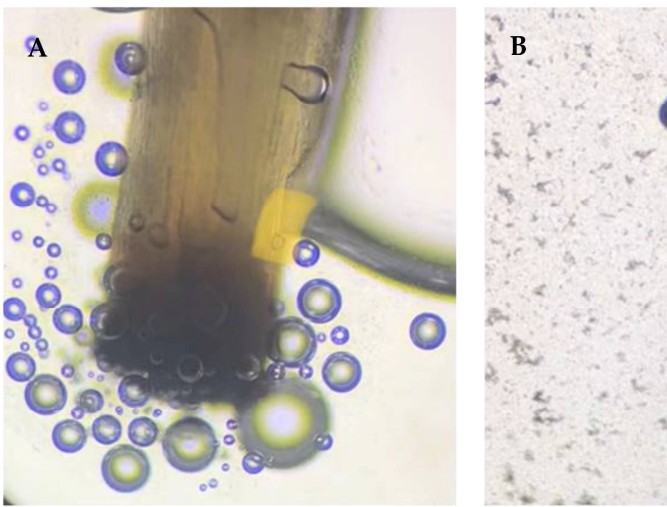
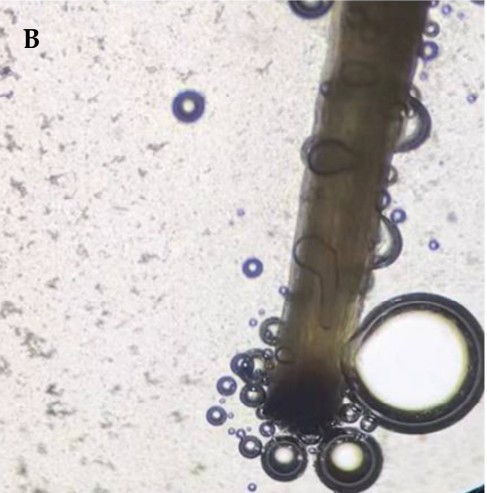

**Figure 1.** Comparison of stigma acceptability of *M. shizongensis* at 10 a.m. (**A**) and 4 p.m. (**B**).

3.2.2. Stigma Fluorescence Detection

After the stigma was artificially pollinated, the pollen tube germination was observed for 72 h. Twelve hours after artificial pollination, more pollen grains were attached to the stigma and almost no pollen tubes had germinated in *M. shizongensis* (Figure 2A). Twenty-four hours after artificial pollination, more pollen grains were attached to the stigma too. Pollen begins to sprout (Figure 2B). Thirty-six hours after artificial pollination, reduced pollen grains attached to the stigma, and more pollen tubes had germinated (Figure 2C). Forty-eight hours after artificial pollination, fewer pollen grains were attached to the stigma, and less pollen tubes had germinated and elongated (Figure 2D). Sixty hours after artificial pollination, the germination of pollen grains decreased (Figure 2E). Seventy-two hours after artificial pollination, the pollen tubes had extended to the base of the style (Figure 2F,G). Less *M. shizongensis* pollen was found to have entered the base of the style compared to the Fuji pollen (Figure 2F). These results demonstrated that the stigma of *M. shizongensis* behaved normally and that its pollen tubes could germinate in the stigma. However, the number of pollen tubes entering the ovary was much lower than that of the Fuji cultivar.

For Fuji, as a high-yielding main cultivar, the germination of pollen in stigma is obvious, but the observation shows that the germination of *M. shizongensis* is much lower than that of Fuji, which indicates that for *M. shizongensis* the amount of pollen entering the ovary is very low.

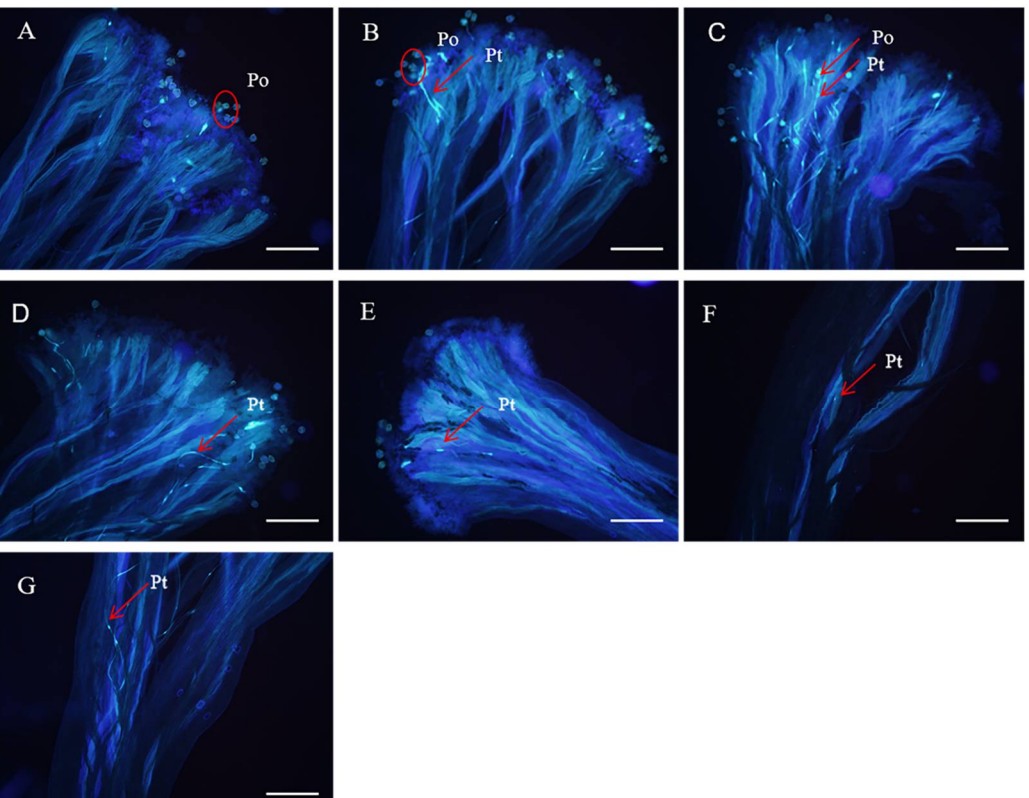

**Figure 2.** Stigma fluorescence in *M. shizongensis*. Pt: pollen tube; Po: pollen. Note: the bar represents 20 µm, (**A**–**G**) represents the germination of pollen at different times after pollination. (**A**) Stigma of crabapple from *M. shizongensis* 12 h after artificial pollination. (**B**) Stigma of crabapple from *M. shizongensis* 24 h after artificial pollination, with pollen grains attached to stigmas. (**C**) Stigma of crabapple from *M. shizongensis* 36 h after artificial pollination. (**D**) Stigma of crabapple from *M. shizongensis* 48 h after artificial pollination. (**E**) Stigma of crabapple from *M. shizongensis* 60 h after artificial pollination. (**F**) Stigma of crabapple from *M. shizongensis* 72 h after artificial pollination. (**G**) Stigma of crabapple from Fuji 72 h after artificial pollination.

3.2.3. Embryological Observation of *M. shizongensis*

Based on the examination of the longitudinal section of the ovary, the ovule primordium of *M. shizongensis* appeared at the initial blooming stage of the flower bud (Figure 3A). As the ovary developed, the peripheral cells of the base of the nucleus divided rapidly and gradually expanded upward to surround the nucleus which is called the integument. Under the epidermis of the nucleus cells, a large cell with dense cytoplasm, a large nucleus, and abundant organelles that was obviously different from the surrounding cells developed into a primordial cell (Figure 3B). This cell then developed into a megaspore mother cell (Figure 3C). The cell division of both the inner and outer integument accelerated over time, with the inner integument growth increasing faster than the outer. The outer integules grew and developed faster than the nucellus cells, so the inner and outer integules were surrounded by the nucellus cells, forming an inverted ovule. Part of the inner and outer integument was unconsolidated above the nucleus cells, forming the micropyle, through which the pollen tube entered the ovule. As the embryo sac mother cell developed, it underwent meiosis to form a tetrad (Figure 3D). However, some embryos aborted before flowering (Figure 3F). Occasionally, when a megaspore was observed, three

cells near the micropyle end were aborted during the tetrad phase, while only cells near the chalaza end continued to develop into a functional megaspore (Figure 3E), which was formed into a binuclear embryo sac (Figure 3G) and a four-nuclear embryo sac (Figure 3H) via mitosis. This analysis showed that during development, the tetrad of *M. shizongensis* disintegrated, leading to the termination of sexual reproduction.

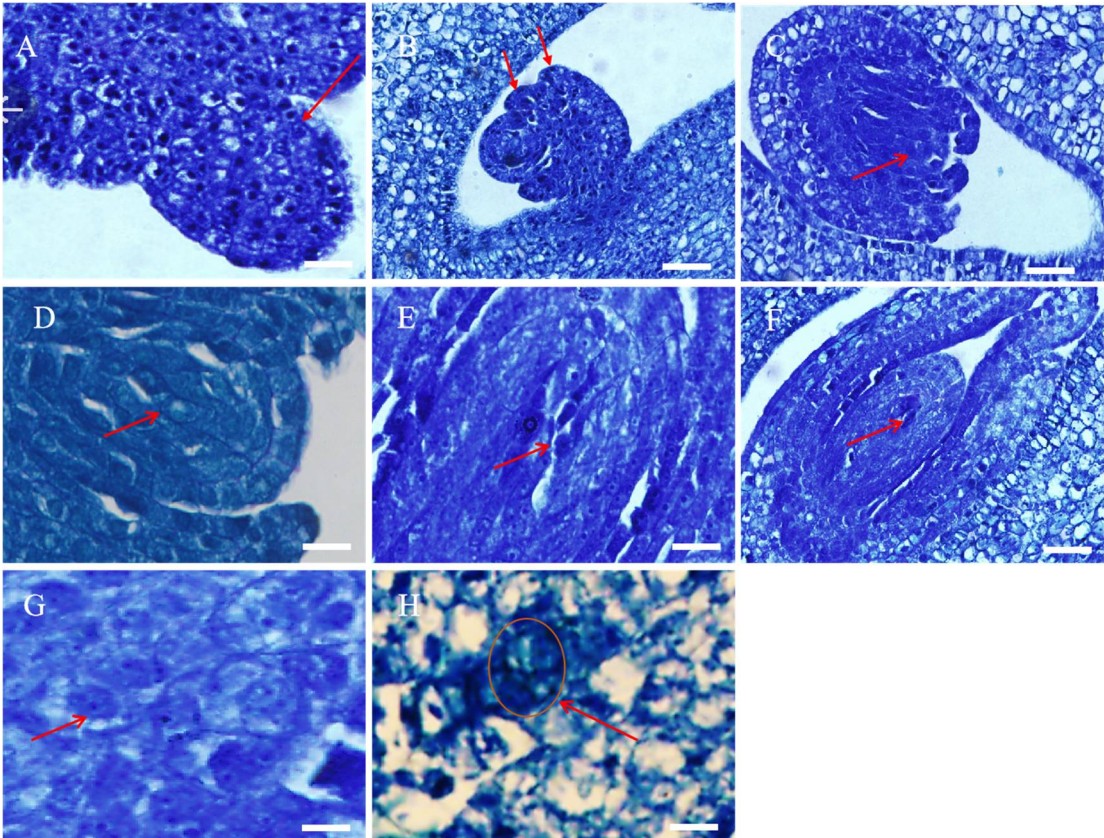

**Figure 3.** Sexual reproductive abortion in *M. shizongensis*. Note: bars represent 20 μm, (**A–H**) represents the developmental stage of embryo sac development. (**A**) The primordium of the embryo sac (denoted with arrows); (**B**) ovule growth and the emergence of primordial cells, the arrow points to integument; (**C**) the megaspore mother cell, with an arrow pointing to the megaspore mother cell; (**D**) tetrad cells; (**E**) the three spores at the end of the foramen degenerated while the chalaza spores developed; (**F**) all tetrads aborted; (**G**) binuclear embryo sac; and (**H**) four-nuclear embryo sac.

*3.3. Further Exploring the Causes of Anther Dysplasia and Male Gamete Development Failure in M. shizongensis*

3.3.1. Anther Morphological Observation of *M. shizongensis*

In the field, about 30 flowers were observed, and the state of each flower is shown in Figure 4: Figure 4A shows the anther condition of *M. shizongensis* at the full flowering stage and Figure 4B shows the anther morphology of the normal variety Fuji at the full flowering stage. By comparing the anther morphology of *M. shizongensis* at the full flowering stage to other normal cultivars, we found that most of the anthers of *M. shizongensis* were immature, small, green and dry, with no signs of cracking. The anthers of Fuji were fuller, yellow in color and split open upon reaching maturity. The process of pollination and fertilization can be completed by wind or insect vectors, but the lack of anther opening in *M. shizongensis* prevents these processes from occurring and affect the process of pollination and fertilization.

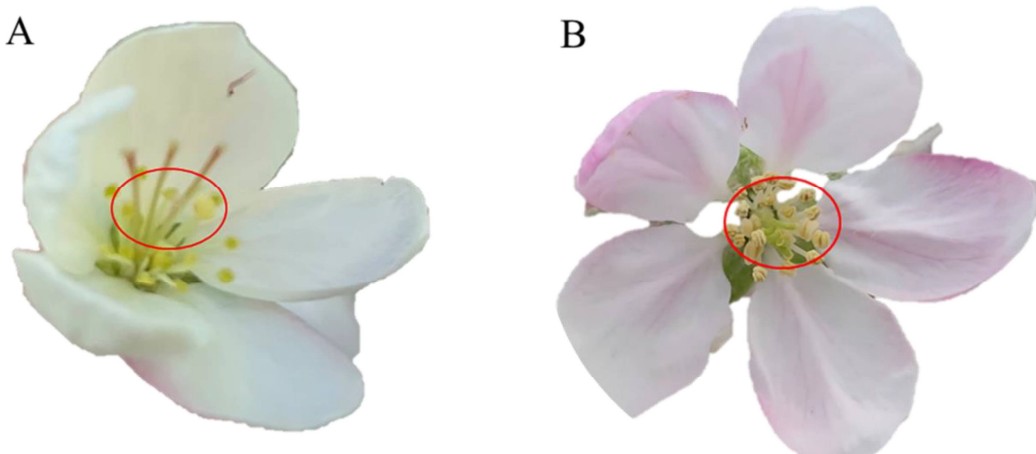

**Figure 4.** Morphological observation of anther in *M. shizongensis* (**A**) and Fuji (**B**). Note: The red circle in the figure is the anther state during the full flowering period.

### 3.3.2. Three-Dimensional Structure of Anthers of *M. shizongensis*

Morphological observations indicated that the anther development of *M. shizongensis* differed from other cultivars, leading us to examine this process more closely with a scanning electron microscope at multiple different resolutions (Figure 5). The selected 10 anthers were analyzed: we observed that the anther development of *M. shizongensis* was highly variable and that the furrows of each anther were uneven. The cracking depth was variable, with most anthers having a deformed, shriveled appearance. No anthers were found to have broken open, a process that is required for pollen shed. The rate of anther deformity reached 100%.

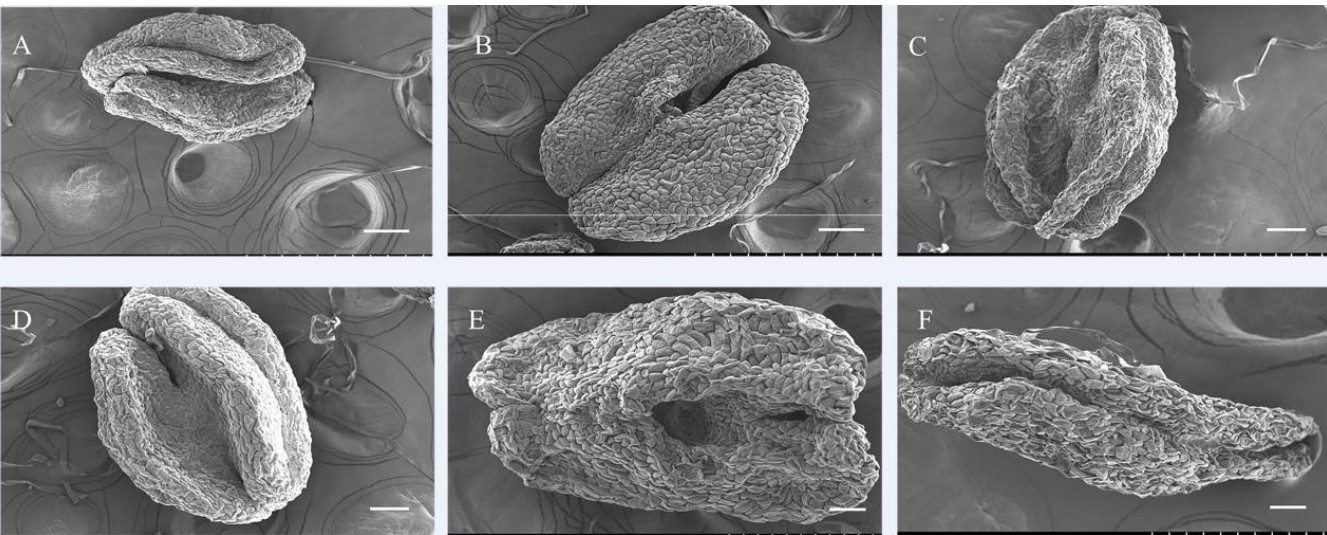

**Figure 5.** Anther stereoscopic structure in *M. shizongensis*. Note: The scales in (**A**,**B**) represent 400 µm. The scales in (**C**,**D**) represent 300 µm, and the scales in (**E**,**F**) represent 200 µm.

### 3.3.3. Development of the Male Gametophyte in *M. shizongensis*

Morphological observation and microscopy indicated that the anthers of *M. shizongensis* had dysplasia, prompting us to develop paraffin sections from them for additional analysis (Figure 6). The anthers were composed of an epidermis, a fiber layer, a middle layer and a tapetum, with visible mature pollen mother cells (Figure 6A). Tetrad was surrounded by a callose wall (Figure 6B) and the tapetum secreted callose in the callosity wall of the tetrad (Figure 6C). After more development, the pollen mother cell and tetrad

were surrounded by tapetum cells and the tetrad further degraded (Figure 6D). Thereafter, the pollen grains were all degraded (Figure 6E); at the same time, it can be observed that the wall of the anthers chamber thickens (Figure 6F). At this point, the tapetum cells began to disappear (Figure 6G), leaving only empty anthers (Figure 6H). Under normal circumstances, when the wall of the anthers chamber thickens, streak-like thickening does not occur at the junction of two pollen sacs on the same side, and the wall is kept thin so that the pollen grains can split the thin wall and disperse at maturity. However, observation of the connection between two similar pollen sacs in *M. shizongensis* showed that, when the wall of the medicine chamber thickened, the connection of the pollen sacs on the same side had different degrees of thickening. This resulted in an inability of the pollen grains to disperse and the eventual abortion of male gametes.

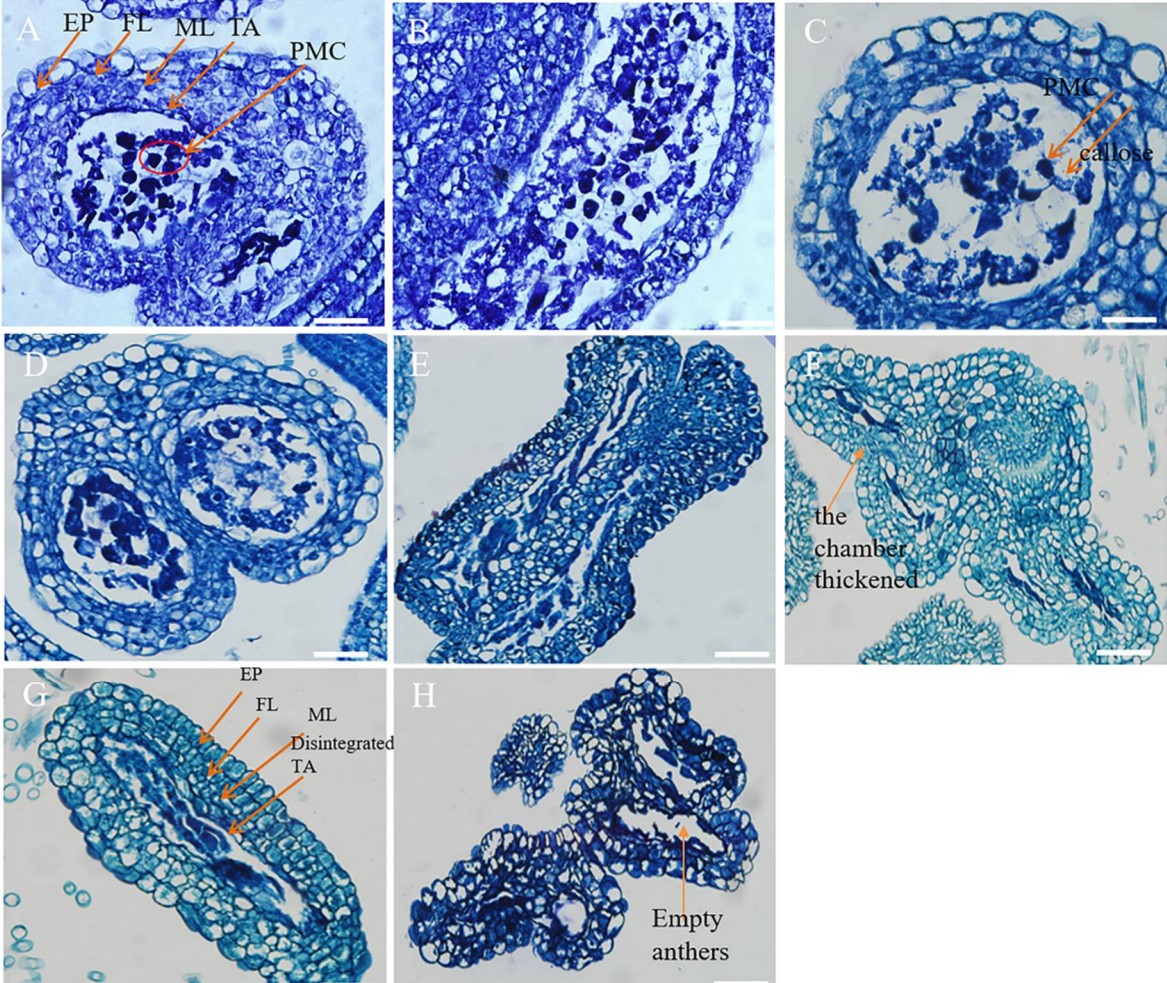

**Figure 6.** Male gametophyte sterilization of *M. shizongensis*. EP: epidermis; FL: fibrous layer; ML: middle layer; TA: tapetum; PMC: pollen mother cell. Note: The bar represents 20 μm. (**A–H**) represents the different stages of male gamete development. (**A**): morphologically normal pollen mother cell; (**B**): tetrad surrounded by callose wall; (**C**): the tapetum secreted callose in the callosity wall of the tetrad; (**D**): tetrad was further degraded; (**E**): the pollen grains were all degraded; (**F**): the wall of the anthers chamber thickens; (**G**): the tapetum cells began to disappear; and (**H**): empty anthers.

## 4. Discussion

Apomixis is a unique type of asexual reproduction which preserves maternal genotypes without meiosis and fertilization to generate clonal offspring [19]. This process has the potential to fix heterosis, which has large implications for breeding high-yielding and stable germplasm [20].

### 4.1. Effects of Different Treatments on Fruit Setting Rate and Satiety Rate

Removal of stamens, followed by bagging, can be carried out at the budding stage to determine if a fruit tree species has apomictic ability. In this work, removal of stamens followed by bagging (DSB) resulted in a 90.7% fruit setting rate, compared to a 78.7% rate after emasculation and bagging (DMB); it is preliminarily proved that *M. shizongensis* has a high rate of apomixis. Xiong [21] previously showed that the apomictic reproductive rate of plants subjected to stigma decapitation had a much higher reproductive rate than those which were fully emasculated, which is in keeping with our results. This disparity may have arisen because *M. shizongensis* has smaller flowers that are prone to damage during emasculation, which may result in flower organ drop. Stigma removal is easier to carry out compared to emasculation and is also less sensitive to variations in the environment. Our results indicated that the 1000-seed weight of pollinated fruits was significantly higher than that under other treatments. This may be caused by a lack of pistil stimulation during apomictic reproduction, as well as a lower level of plant hormones, which negatively impacts fruit development.

### 4.2. Effects of Male Gamete Development on Apomixes

In most species, the dehiscence of anthers and transmission of pollen during the flowering period are required to complete the process of pollination and fertilization. Liu [22] showed that the embryo sac development of apomictic plants from the genus *M. hupehensis* var. *pingyiensis* is delayed. This causes the embryo to not be fully developed during the reproductive period of the male gamete, resulting in apomixis. In this study, the anthers of *M. shizongensis* were not mature at the full flowering stage. Microscopic examination of anthers showed that they were completely closed at the full flowering stage and had no pollen bursting. Also, paraffin sectioning revealed abnormal development of the tapetum, which plays a critical role during the development of pollen grains. When tapetum cells transport nutrients to the anther compartment, the tapetum secretes callose, which can disintegrate the callosity wall of the pollen mother cell and tetrad, thus separating the young mononuclear pollen grains from each other and ensuring normal development. We observed that abnormal tapetum development resulted in the encapsulation of pollen grains, leading to pollen abortion. Under normal circumstances, when the pollen matures, the pollen sac opens as the chamber wall thickens. After flowering, when anthers extend out of the flower, the cells lose water and the tension caused by the thickening of the cell wall of the fiber layer will split the weakest part of the connection of the pollen sac on the same side, and the pollen grains will escape through the crack. However, we found that the anthers of *M. shizongensis* had striate thickening at the junction of the two pollen sacs, so the anthers could not split when water was lost during flowering, resulting in an inability of pollen to disperse. Therefore, we speculated that anther malformation and pollen grain dysplasia led to apomixis in *M. shizongensis*. In an earlier study on apomixis by Maia, pollen absence was found to be closely related to apomixis [23]. In this study, to test this hypothesis, Maia studied breeding systems of 16 species of Melastomataceae, and the results indicate that plants with low pollen vigor or no pollen vitality are more likely to produce strictly apomictic fruits. However, the genetic backgrounds of plants are complex, and it remains an open question whether apomixis in *M. shizongensis* is caused by the dysplasia of male gametes.

### 4.3. Pollen Germination on Stigma of M. shizongensis after Pollination

Stigmas are the site of pollen grain formation and germination, which play important roles in sexual reproduction. Previously, Ossama showed that, in the Amygdaloideae subfamily, pollination within two days after emasculation resulted in fruit setting rates of 34.02% to 49.98%, while later pollination significantly reduced the fruit setting rate [24]. In this work, pollination was carried out immediately after emasculation and stigma receptivity tests indicated a normal level of receptivity under all conditions. However, after pollination, the pollen germination took place later than that of *M. hupehensis* var.

*pingyiensis*. Seventy-two hours after pollination, some pollen entered the bottom of the style, but the amount was less than that of normal cultivars of apple. Jahed reported that 'Honey Crisp' and 'Golden Delicious' pollen tubes form optimally after 48 h, while 'Delicious' and 'Ralph Shay' tubes have ideal lengths at 96 h. In Fuji and 'Gala', regardless of the pollen source, it takes up to 72 h for optimal pollen tube growth [25]. Additionally, Heilmann [26] found that the time from pollen landing on stigma to fertilization varied significantly across plant species and varieties. Li [27] observed the pollen tubes of 'Red Star' apple after pollination, and found that pollen tubes began to germinate at the upper part of the style 24 h after pollination, with a dramatic increase in germination after 48 h. We found that 24 h after pollination, no pollen tubes had germinated at the stigmas of *M. shizongensis*. However, the number of pollen tubes that had germinated after 72 h was significantly lower than the normal species. There are research findings suggest that pollination may therefore fail if the pollen reaches the pistil after a delay of longer than 3 h [28]. This means that the amount or time of pollen entering the ovary will affect the development of the embryo. In addition, the ovules are still rudimentary or immature at the time of pollination [29]. We consider a competitive advantage that may help to compete for pollination, even when the female gametes themselves have not been formed. So when pollination fails, this mechanism leads the embryo sac to another mode of development.

### 4.4. Sexual Embryo Sac Abortion of M. shizongensis

The reproductive strategies employed by species have significant impacts on their evolution. When normal sexual development is blocked, apomictic reproduction becomes the primary strategy to avoid sterility. In the life cycle of sexual organisms, a specialized cell division—meiosis—reduces the number of chromosomes from two sets (2n, diploid) to one set (n, haploid), while fertilization restores the original chromosome number. In contrast, mitosis produces two identical daughter cells. The replacement of meiosis by mitosis is a key component of apomixis [30]. During the tetrad period, nonsister chromatids exchange in fragments, genetic material between the chromosomes of the parent, and the abortion of the tetrad period ensures that the genetic information is completely consistent with the mother.

Apomixis in apple is mainly derived from ansporogenesis. In normal sexual reproduction, the megaspore mother cell undergoes meiosis to form a functional megaspore. This megaspore develops into a binuclear embryo sac, four-nuclear embryo sac and eventually eight-nuclear embryo sac. However, observation of paraffin sections of the ovary of *M. shizongensis* showed that while some megaspore mother cells could develop normally, most aborted in the tetrad stage and failed to undergo normal meiosis. Instead, the nucleus cells underwent mitosis to form the embryo sac and then differentiated into an embryo. This was consistent with other results, which indicated that most megaspore mother cells of *M. hupehensis* var. *pingyiensis* degenerated in the tetrad and could not carry out normal sexual reproduction, resulting in apomixis. Previous studies have shown that the embryo sac development of *M. hupehensis* var. *pingyiensis* is divided into two parts: sexual embryo sac development and abortion, and asexual embryo sac development. Although reproduction is predicted to be similar in *M. shizongensis*, we found that only the abortion of the first part of the sexual embryo sac occurred. Additional research is therefore needed to better understand the development of the asexual embryo sac.

### 4.5. Apomixis Type of M. shizongensis

Apomixis can be divided into two different types according to its underlying cause and embryological characteristics. Based on the underlying cause, it can be categorized as either sporogenous or gametophytic [31]. In this study, observation of the development of female gametophytes revealed that although megaspore mother cells of *M. shizongensis* could undergo meiosis, they all degenerated with the abortion of the tetrad, indicating that gametophytic apomixis was taking place. This same type of apomixis has been found in other apple species, such as *M. hupehensis* var. *pingyiensis* and *M. toringoides*

*Hughes*. However, only sexual abortion was observed in this study, and further observation should be carried out on the subsequent development processes to determine the specific type of gametophyte apomixis. Apomixis can also be categorized as either facultative or obligate [8]. Apomixis species that lose the ability to reproduce sexually and can only produce offspring through apomixis are considered to undergo obligate apomixis, while species that can still sexually reproduce are considered to have facultative apomixis. During reproductive development, sexual reproduction and apomixis do not exist simultaneously, making it impossible to know which process has generated seeds under natural conditions. Our observation of stigma fluorescence identified a small number of pollen tubes that penetrated the ovary, meaning some level of sexual reproduction could occur. Observation of female gametes indicated that most tetrads were aborted, although few continued to develop into two nuclear embryo sacs and four nuclear embryo sacs. Based on these results, *M. shizongensis* may undergo facultative apomixis, which is consistent with several other apple species. However, it is still possible that the male gamete does not combine with the female gamete after entering the ovary and *M. shizongensis* could still be categorized as undergoing obligate apomixis. If it is obligate apomixis, the offspring will show complete consistency, which will greatly shorten the age of subsequent stock breeding. If it is facultative apomixis, the hybrid offspring should be tested first in the breeding process to improve breeding. Additional observation of the reproductive process is therefore needed to definitively categorize apomixis in *M. shizongensis*.

## 5. Conclusions

In this study, the *M. shizongensis* was bagged and its stigmas and males were removed. *M. shizongensis* was found to still set fruit without pollination, which demonstrated that it could undergo apomixis. The apomictic potential of *M. shizongensis* was found to have three underlying causes. Firstly, its anthers have deformed structures due to improper disintegration of the tapetum layer that prevents normal pollen grain formation. Secondly, the stigma of the begonia was normal, but only a few pollen tubes were found to enter the ovary. Thirdly, some embryo sacs of *M. shizongensis* were aborted during meiosis of the tetrad, leading to the termination of sexual reproduction. *M. shizongensis* has apomictic properties. This means that the offspring is exactly the same as the mother, and their offspring can be bred with real seeds, which can greatly reduce the possibility of seedlings carrying the virus. Apomixis is the physiological basis for the reproduction of self-root anvil seedlings. The study of apomixis characteristics of *M. shizongensis* can provide more choices for apple stock breeding and create more stock varieties that meet the needs of the current development on the basis of the existing resources.

**Author Contributions:** Conceptualization, Z.L. and Y.F.; methodology, Y.F.; software, Y.F.; validation, Y.F.; formal analysis, Y.F. and R.N.; data curation, Y.F., Z.W., Y.H. (Ying He) and Y.H. (Yu Hu); writing—original draft preparation, Y.F.; writing—review and editing, L.S.; funding acquisition, Z.L. All authors have read and agreed to the published version of the manuscript.

**Funding:** This research was funded by the Modern Agro-industry Technology Research System of China (CARS-27).

**Data Availability Statement:** Data are contained within the article.

**Acknowledgments:** We thank the Horticulture Science Research Center of the College of Horticulture, NWAFU for their technical support in this work.

**Conflicts of Interest:** The authors declare no conflict of interest.

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
