# Peer review of "A Study on the Causes of Apomixis in Malus shizongensis"

_horticulturae, doi:10.3390/horticulturae9080926_

Round 1

Reviewer 1 Report

Reviewer response to author:

Abstract:

1. The results presented in the abstract are straightforward and provide insights into the potential causes of apomixis in Malus shizongensis. However, the abstract lacks specific data, such as percentage values or statistical significance, to support the conclusions made in the study. Adding some quantitative information would strengthen the abstract's impact.  On the other side, the abstract mentions the occurrence of anther dysplasia and male gamete development failure as the primary causes of apomixis, it would be valuable to elaborate on the implications of these findings and how they contribute to understanding the unique reproductive process in Malus shizongensis.

Introduction:

1.      Overall, the introduction provides a good background on apomixis and its occurrence in different plant species, but it could benefit from some improvements to enhance the coherence and relevance to the current study on Malus shizongensis' apomixis. The introduction part provides an overview of apomixis, a unique reproductive process that produces fertile offspring without the combination of sperm and egg cells. The paper highlights the importance of studying apomixis in plants and provides a brief history of research on apomixis in different plant species. The introduction section also mentions the previous research on apomixis in Malus hupehensis Rehd. gametophytic and introduces Malus shizongensis Liu sp. Nov, a newly discovered species closely related to Malus hupehensis, is the focus of the study.

2.      The transition from discussing research in various plant species to focusing on Malus is abrupt. A smoother transition, possibly by providing more context on Malus species and their relevance to apomixis research, would make the flow of the introduction more coherent.

3.      The last sentence of the introduction states that the current study examined the development of Malus shizongensis' pistils, stamens, and embryo sacs to understand the causes of apomixis in this species. It would be helpful to provide a brief overview of the methodologies used in this study to investigate apomixes, such as paraffin sectioning, electron scanning microscopy, or any other techniques employed. It would be beneficial to expand on this point and discuss the potential applications or implications of the study's findings for breeding programs or other areas of research.

Results:

The Results section is well-structured and presents the findings in a clear manner. However, it would be helpful to include subsection headings for each of the findings to enhance readability and organization. Exploring the potential of apomixis for crop improvement in Malus shizongensis and other species? Investigating the role of plant hormones in apomictic reproduction and fruit development? Further exploring the causes of anther dysplasia and male gamete development failure in Malus shizongensis?

1.      The tables present essential data related to fruit setting rates and fruit characteristics. However, it would be helpful to include information on the statistical analysis performed, such as the type of statistical tests used and the significance levels.

2.      In Figure 1, the legend should be more descriptive, specifying the exact time of the morning and afternoon measurements for stigma acceptability. This would provide readers with a better understanding of the experimental setup.

3.      For Figure 2, additional information could be provided about the significance of the observation of Malus shizongensis pollen's lower entry into the base of the style compared to Fuji pollen. Further analysis or explanation of the implications of this difference would enhance the interpretation of the results.

4.      In Figure 3, the labeling of the different stages of sexual reproductive abortion in Malus shizongensis could be improved to make it clearer and easier to follow. Consider using sequential letters or numbers for each stage.

5.      In Figure 4, it is not explicitly stated how many anthers were observed for each condition (Malus shizongensis and Fuji). Adding the sample size for each observation would provide valuable context for the results.

6.      In Figure 5, additional information on the total number of anthers examined and the percentage of anthers displaying dysplasia in Malus shizongensis would be helpful to understand the prevalence of this characteristic.

7.      The paraffin sections in Figure 6 could benefit from clearer labeling of the different stages in the development of the male gametophyte. Ensure that the labeling is consistent with the corresponding descriptions in the text.

8.      The results are presented in a descriptive manner, but additional statistical analysis or comparisons between treatments could further strengthen the findings. Consider conducting statistical tests to assess significant differences between treatments and highlight them in the text or tables.

Discussion:

The discussions section provides a comprehensive analysis of the findings. However, it would be beneficial to start the section with a brief introduction or summary of the main results to set the context for the discussion. Throughout the Discussions section, consider incorporating references to relevant literature or previous studies that support the interpretations and conclusions presented. This will add credibility and depth to the discussion. Overall, the discussions section provides a thorough examination of the results. By addressing the above suggestions and including references to support the findings, the section will become even more informative and robust.

1.      In the subsection "4.1. Effects of different treatments on fruit setting rate and satiety rate," it would be helpful to clarify the specific treatments associated with T1, T2, and T3. This information is essential for readers to understand the comparisons being made.

2.      In the subsection "4.2. Effects of male gamete development on apomixes," the discussion of anther malformation and pollen grain dysplasia as a cause of apomixis is insightful. However, it could be strengthened by referring to specific research or studies that support this conclusion.

3.      In the subsection "4.3. Pollen germination on the stigma of Malus shizongensis after pollination," while the findings are discussed in detail, it would be helpful to include the implications of delayed pollen germination and lower pollen entry into the style for the overall apomixis process.

4.      In the subsection "4.4. Sexual embryo sac abortion of Malus shizongensis," the discussion on the observations of sexual embryo sac abortion is well-presented. To further enhance the discussion, consider providing possible explanations or hypotheses for why most megaspore mother cells undergo abortion at the tetrad stage.

5.      In the subsection "4.5. Apomixis type of Malus shizongensis," the distinction between facultative and obligate apomixis is well-explained. However, it would be valuable to discuss the implications of these categories on the reproductive strategy and potential breeding applications of Malus shizongensis.

6.      In the "Conclusions" section (subsection "5. Conclusions"), the summary of the apomictic potential of Malus shizongensis is clear and well-organized. However, it could be further expanded to include the broader implications of these findings in the context of plant breeding, horticulture, or agricultural practices.

References:

The references are properly formatted in terms of citation style. Several references are cited multiple times, which might be due to duplicate entries. Please review the reference list carefully and eliminate any repetitive entries. However, there are some duplicates in the reference list (e.g., references 4/5 and 9). Please remove the duplicate references to avoid redundancy.

In the introduction part line 35, Cheng et Jiang?

If possible, consider adding a few more recent references, to provide readers with the latest research developments in the field. It would be helpful to provide more diversity in the sources by including studies from different research groups or regions. This will enhance the credibility and validity of the findings presented in the present paper.

Author Response

Response to Reviewer 1 Comments

Dear experts, thank you for your patience. At present, we have revised these opinions one by one according to your opinions, and marked them red in the article, and replied one by one below.

Abstract:

  1. The results presented in the abstract are straightforward and provide insights into the potential causes of apomixis in Malus shizongensis. However, the abstract lacks specific data, such as percentage values or statistical significance, to support the conclusions made in the study. Adding some quantitative information would strengthen the abstract's impact.  On the other side, the abstract mentions the occurrence of anther dysplasia and male gamete development failure as the primary causes of apomixis, it would be valuable to elaborate on the implications of these findings and how they contribute to understanding the unique reproductive process in Malus shizongensis.

Response : Thanks for the suggestions of the review experts. At present, I have modified the summary part for your suggestions, and marked the modified part in red.

Introduction:

  1. Overall, the introduction provides a good background on apomixis and its occurrence in different plant species, but it could benefit from some improvements to enhance the coherence and relevance to the current study on Malus shizongensis'apomixis. The introduction part provides an overview of apomixis, a unique reproductive process that produces fertile offspring without the combination of sperm and egg cells. The paper highlights the importance of studying apomixis in plants and provides a brief history of research on apomixis in different plant species. The introduction section also mentions the previous research on apomixis in Malus hupehensis Rehd. gametophytic and introduces Malus shizongensis Liu sp. Nov, a newly discovered species closely related to Malus hupehensis, is the focus of the study.
  2.  The transition from discussing research in various plant species to focusing on Malus is abrupt. A smoother transition, possibly by providing more context on Malus species and their relevance to apomixis research, would make the flow of the introduction more coherent.

Response 2: We have adjusted the logic of the introduction to ensure that there is a smooth transition that makes the reader more coherent

3      The last sentence of the introduction states that the current study examined the development of Malus shizongensis' pistils, stamens, and embryo sacs to understand the causes of apomixis in this species. It would be helpful to provide a brief overview of the methodologies used in this study to investigate apomixes, such as paraffin sectioning, electron scanning microscopy, or any other techniques employed. It would be beneficial to expand on this point and discuss the potential applications or implications of the study's findings for breeding programs or other areas of research.

Response 3: Thank you very much for your valuable advice. I have added experimental methods in the introduction section and extended the significance of this study for breeding, specific in line 76 -77and 78-82.

Results:

The Results section is well-structured and presents the findings in a clear manner. However, it would be helpful to include subsection headings for each of the findings to enhance readability and organization. Exploring the potential of apomixis for crop improvement in Malus shizongensis and other species? Investigating the role of plant hormones in apomictic reproduction and fruit development? Further exploring the causes of anther dysplasia and male gamete development failure in Malus shizongensis?

Response : Thank you very much for your detailed comments. At present, I have revised them in detail according to your several comments. I have divided the results section according to your proposa, and added subsection headings. 

  1.  The tables present essential data related to fruit setting rates and fruit characteristics. However, it would be helpful to include information on the statistical analysis performed, such as the type of statistical tests used and the significance levels.

Response 1: The type of statistical analysis and the significance levels has been indicated in the annotation below the table.

  1.   In Figure 1, the legend should be more descriptive, specifying the exact time of the morning and afternoon measurements for stigma acceptability. This would provide readers with a better understanding of the experimental setup.

Response 2: In the description of Figure 1, I have added the exact time, in line 205-206, and also described in detail in the materials and methods.

3      For Figure 2, additional information could be provided about the significance of the observation of Malus shizongensis pollen's lower entry into the base of the style compared to Fuji pollen. Further analysis or explanation of the implications of this difference would enhance the interpretation of the results.

Response 3: Thank you for your comments. I have explained and analyzed this part in the article, and  marked it in red.

4      In Figure 3, the labeling of the different stages of sexual reproductive abortion in Malus shizongensis could be improved to make it clearer and easier to follow. Consider using sequential letters or numbers for each stage.

Response 4: I have marked the different stages in the text Figures

  1. 5.   In Figure 4, it is not explicitly stated how many anthers were observed for each condition (Malus shizongensis and Fuji). Adding the sample size for each observation would provide valuable context for the results.

Response 5: At present, I have quantified the observed sample size in the text.

6  In Figure 5, additional information on the total number of anthers examined and the percentage of anthers displaying dysplasia in Malus shizongensis would be helpful to understand the prevalence of this characteristic.

Response 6: The modification of this part is mainly reflected in two parts in the text. First, in 2.2.5, the operation of this part is explained in more detail, hoping to provide readers with good reading experience; secondly, in 3.3.2of the results part,the observed samples were quantified.

7      The paraffin sections in Figure 6 could benefit from clearer labeling of the different stages in the development of the male gametophyte. Ensure that the labeling is consistent with the corresponding descriptions in the text.

Response 7: I have marked part of the structure in Figure 6, hope to better express the meaning of the graph.

8      The results are presented in a descriptive manner, but additional statistical analysis or comparisons between treatments could further strengthen the findings. Consider conducting statistical tests to assess significant differences between treatments and highlight them in the text or tables.

Response 8: Thank you very much for your comments. At present, I have added in the paper of the significant differences between the description, mainly reflected in 3.1, 3.2.1 and 3.2.2 part, because the test only for this test material, and for other varieties of apomixis related research has been very thorough, so in the male gamete development and embryo development part no comparative analysis, only studied Malus shizongensis, but I will learn your advice in the future research, design more logical test.

Discussion:

The discussions section provides a comprehensive analysis of the findings. However, it would be beneficial to start the section with a brief introduction or summary of the main results to set the context for the discussion. Throughout the Discussions section, consider incorporating references to relevant literature or previous studies that support the interpretations and conclusions presented. This will add credibility and depth to the discussion. Overall, the discussions section provides a thorough examination of the results. By addressing the above suggestions and including references to support the findings, the section will become even more informative and robust.

  1.     In the subsection "4.1. Effects of different treatments on fruit setting rate and satiety rate," it would be helpful to clarify the specific treatments associated with T1, T2, and T3. This information is essential for readers to understand the comparisons being made.

Response 1: Thank you very much for your valuable comments. We have revised your comments one by one in the discussion section. In 4.1, we have connected the processing with the previous T1, T2, T3, hoping to help readers to read easily.

  1.   In the subsection "4.2. Effects of male gamete development on apomixes," the discussion of anther malformation and pollen grain dysplasia as a cause of apomixis is insightful. However, it could be strengthened by referring to specific research or studies that support this conclusion.

Response 2: In 4.2, we have added studies on this conclusion to enhance the conclusion. In line 374-376.

  1.     In the subsection "4.3. Pollen germination on the stigma of Malus shizongensis after pollination," while the findings are discussed in detail, it would be helpful to include the implications of delayed pollen germination and lower pollen entry into the style for the overall apomixis process.

Response 3: I have taken your advice and included the impact of delayed pollen germination and lower pollen entry into the style for the overall apomixis process in 4.3

  1. In the subsection "4.4. Sexual embryo sac abortion of Malus shizongensis," the discussion on the observations of sexual embryo sac abortion is well-presented. To further enhance the discussion, consider providing possible explanations or hypotheses for why most megaspore mother cells undergo abortion at the tetrad stage.

Response 4: In Discussion 4.4, we have added relevant hypotheses for the abortion of megaspore mother cells in the tetrad.

5      In the subsection "4.5. Apomixis type of Malus shizongensis," the distinction between facultative and obligate apomixis is well-explained. However, it would be valuable to discuss the implications of these categories on the reproductive strategy and potential breeding applications of Malus shizongensis.

Response 5: Thank you for your comments, which we have discussed about these categories on the reproductive strategy and potential breeding applications of Malus shizongensis in 4.5.

  1. In the "Conclusions" section (subsection "5. Conclusions"), the summary of the apomictic potential of Malus shizongensis is clear and well-organized. However, it could be further expanded to include the broader implications of these findings in the context of plant breeding, horticulture, or agricultural practices.

Response 6: Dear expert, we have added in the conclusion section about the impact in plant breeding.

References:

The references are properly formatted in terms of citation style. Several references are cited multiple times, which might be due to duplicate entries. Please review the reference list carefully and eliminate any repetitive entries. However, there are some duplicates in the reference list (e.g., references 4/5 and 9). Please remove the duplicate references to avoid redundancy.

Response : Thank you very much for your care. We have revised and checked the references in detail

In the introduction part line 35, Cheng et Jiang?

Response : We have revised it up,in line 35.

Reviewer 2 Report

Apomixis is a method of seed reproduction of angiosperms in which the embryo develops from an unfertilized egg. At its core, it is nothing more than a natural cloning mechanism. Manipulations with it and use in breeding open up opportunities for preserving valuable genotypes in a number of generations, creating non-splitting hybrid forms, fixing heterosis, and, therefore, promise great economic benefits. For this reason, the study of the authors is undeniably relevant.

The study is based on various cytological methods. There are some remarks that need to be noted.

1.     In paragraphs 2.2.3, 2.2.5, the brands of the microscope, the country of manufacture must be indicated.

2.     In paragraph 2.2.3, you must specify filters for a fluorescent microscope.

3.     How appropriate is the use of the term “treatment” in your case of pollination?

4.     The experimental scheme indicates a single pollination. Then it is not clear where the pollen comes from on the stigma after 36 hours of pollination, if after 12 hours of pollination it was not there.

5.     Fig. 2 needs improvement. The arrow shows pollen grains stained with aniline blue, but they are not stained and this may mean that they are not there. Also, in this picture (G, H), the arrows point to pollen tubes, but they are not there.

6.     The photographs in Fig. 3 are not clear.

7.     Figure captions should be expanded, they should be self-sufficient.

8.     In Fig. 6, no callose is visible (should indicate with arrows what the authors wanted to show here), and there is no description of the figure in the caption to it.

9.     Line 283. The species name should be in italics.

Based on the mentioned above, I think that this article can by recommended for publication in the « Horticulturae» after revision.           

Author Response

Response to Reviewer 2 Comments

Dear experts, thank you for your valuable comments. I have revised the manuscript according to your comments, marking the revised part in red, and giving a reply below.

Apomixis is a method of seed reproduction of angiosperms in which the embryo develops from an unfertilized egg. At its core, it is nothing more than a natural cloning mechanism. Manipulations with it and use in breeding open up opportunities for preserving valuable genotypes in a number of generations, creating non-splitting hybrid forms, fixing heterosis, and, therefore, promise great economic benefits. For this reason, the study of the authors is undeniably relevant.

The study is based on various cytological methods. There are some remarks that need to be noted.

  1.     In paragraphs 2.2.3, 2.2.5, the brands of the microscope, the country of manufacture must be indicated.

Response 1: I have marked the model number and production country of the microscope in 2.2.3 and 2.24

  1.   In paragraph 2.2.3, you must specify filters for a fluorescent microscope.

Response 2: I have been at 2.2.3, adding the fluorescence microscope for filter color.

  1.     How appropriate is the use of the term “treatment” in your case of pollination?

Response 3: Thank you for this question, first of all, I want to explain to you, in order to preliminary verification of apomixis characteristics, we set up three treatments, one of the processing to pollination, its purpose is to ensure the fruit set rate after pollination, pollination is artificial pollination, rather than natural pollination, so I think pollination is one of my treatments.

  1.  The experimental scheme indicates a single pollination. Then it is not clear where the pollen comes from on the stigma after 36 hours of pollination, if after 12 hours of pollination it was not there.

Response 1: Dear expert, I would like to explain the process of this experiment to you. I hope I can solve your question. The experimental treatment was artificial pollination in the morning, the number of pollinated flowers was sufficient, the timing began after pollination, 3-5 stigmas were collected every 12h, fixed in FAA solution, and sampling was stopped after 72h, in order to observe the germination of pollen.

  1.   2 needs improvement. The arrow shows pollen grains stained with aniline blue, but they are not stained and this may mean that they are not there. Also, in this picture (G, H), the arrows point to pollen tubes, but they are not there.

Response 4: Thank you for your valuable advice. Now I have modified Figure 2. In Figure 2B, because the pollen has not yet germinated in the stigma, you can only see the shape of the pollen, in Figure G, the amount of pollen entering the ovary is small, so the light is very weak. In Figure H, I replaced it, this will make the picture even clearer.

  1.     The photographs in Fig. 3 are not clear.

Response 5: We have adjusted the clarity of Figure 3

  1.   Figure captions should be expanded, they should be self-sufficient.

Response 6: We have taken your advice to extend some of the Figure captions.

  1.    In Fig. 6, no callose is visible (should indicate with arrows what the authors wanted to show here), and there is no description of the figure in the caption to it.

Response 7: Thank you very much for your comments, I have marked part of the structure in Figure 6, hope to better express the meaning of the graph.

  1.    Line 283. The species name should be in italics.

Response 8: I have modified the format

Round 2

Reviewer 2 Report

Dear authors of the article. I thank you for your responses to my review. But I have not received satisfactory explanations and corrections from my point of view.

1.     Line 128, you have indicated as a blue filter, it is not scientific. 365 nm excitation filter and 420 nm emission filter are used to study aniline blue staining.

2.     Lines 129, 157 country of origin Japan is usually capitalized.

3.     After your explanation, I still cannot agree with the term «treatment» for pollination. Treatment with pollen is pollination. And how can there be pollination before pollination?

I propose to use, for example, the classical term «pollination in buds».

4.     I didn't get an answer to my question. The experimental scheme indicates a single pollination. Then it is not clear where the pollen comes from on the stigma after 36 hours of pollination, if after 12 hours of pollination it was not there.

5.     Also, I did not see much change for the better in Fig. 2.

On Fig. 2A there are no markings. What would you like to show here?

The unsprouted pollen grains in Fig. 2B should glow just as brightly as the sprouted ones. Perhaps this is a defect in your coloring.

On Fig. 2E pollen tubes are clearly visible but not marked.

What you have shown with the arrows in Fig. 2G, these are not pollen tubes.

6.     Why do you think that in Fig. 6 B it is callose? What would you like to show in Fig. 6 C, D, F? Designations are required.

 Based on the mentioned above, I think that this article can by recommended for publication in the « Horticulturae» after revision.         

Author Response

Response to Reviewer 2 Comments

Dear experts, thank you for your valuable comments.I was inspired a lot from your questions, I have revised the manuscript according to your comments, marking the revised part in red, and giving a reply below.

  1. Line 128, you have indicated as a blue filter, it is not scientific. 365 nm excitation filter and 420 nm emission filter are used to study aniline blue staining.

Response 1: I have re-supplemented the information on the microscope instrument, Aniline stained material was then visualized using a fluorescence microscope with UV excitation light.

  1.   Lines 129, 157 country of origin Japan is usually capitalized.

Response 2: I have changed the country name to capital

  1. After your explanation, I still cannot agree with the term «treatment» for pollination. Treatment with pollen is pollination. And how can there be pollination before pollination?

I propose to use, for example, the classical term «pollination in buds».

Response 3: Dear expert, I have absorbed your advice,With pollination in buds instead of treatment

  1. I didn't get an answer to my question. The experimental scheme indicates a single pollination. Then it is not clear where the pollen comes from on the stigma after 36 hours of pollination, if after 12 hours of pollination it was not there.

Response 4: I'm sorry that the last explanation failed to answer your question, and I will answer your question again:This experiment was pollinated in buds, and the number of pollinated flowers was sufficient, with large numbers of samples to eliminate inter-individual variation. The stigmas were collected once every 12h after pollination, for example, with twenty pollinated flowers, the stigmas of four flowers were randomly collected into FAA at 12h, also, after 24h of pollination, The stigma of 4 flowers were randomly adopted again, all until sampling was stopped after 72h.The purpose of this experiment is to verify the germination of pollen on the stigma, the elongation of the pollen tube, through the observation of this experiment, to verify whether the amount of pollen entering the ovary is normal. Maybe my original figure 2 misled you, but now I have improved figure 2, hope to solve some of your questions.

  1. Also, I did not see much change for the better in Fig. 2.

On Fig. 2A there are no markings. What would you like to show here?

The unsprouted pollen grains in Fig. 2B should glow just as brightly as the sprouted ones. Perhaps this is a defect in your coloring.

On Fig. 2E pollen tubes are clearly visible but not marked.

What you have shown with the arrows in Fig. 2G, these are not pollen tubes.

Response 5: Dear reviewer, thank you very much for your suggestions. Regarding Figure 2, I was indeed inspired by you. So far, I have changed Figure 2, marked, and revised the elaboration part. Thank you again for your suggestions.I hope that my modification will satisfy you.

  1. Why do you think that in Fig. 6 B it is callose? What would you like to show in Fig. 6 C, D, F? Designations are required.

Response 6: I have annotated each stage in the notes of Figure 6 and described each figure in the main text. Callose is secreted by the tapetum, so it is located closest to the tapetum. Callose surrounds the tetrad, causing lysis of the male gametophyte, So not only the part of the arrow, but the other filaments that surround the pollen grains are callose.
